# Are Differences in Inflammatory Markers between Patients with and without Hypertension-Mediated Organ Damage Influenced by Circadian Blood Pressure Abnormalities?

**DOI:** 10.3390/jcm11051252

**Published:** 2022-02-25

**Authors:** Nestor Vazquez-Agra, Ana-Teresa Marques-Afonso, Anton Cruces-Sande, Ignacio Novo-Veleiro, Jose-Enrique Lopez-Paz, Antonio Pose-Reino, Alvaro Hermida-Ameijeiras

**Affiliations:** 1Department of Internal Medicine, University Hospital of Santiago de Compostela, 15706 A Coruña, Spain; ana.teresa.marques.afonso@sergas.es (A.-T.M.-A.); ignacio.novo.veleiro@gmail.com (I.N.-V.); joseenriquelopezpaz@yahoo.es (J.-E.L.-P.); antonio.pose.reino@sergas.es (A.P.-R.); alvaro.hermida@usc.es (A.H.-A.); 2Department of Biochemistry and Molecular Biology, University of Santiago de Compostela, 15782 A Coruña, Spain

**Keywords:** blood pressure, arterial hypertension, inflammatory markers, hypertension-mediated organ damage

## Abstract

We aimed to explore the influence that the circadian blood pressure (BP) profile could exert on the correlation between some inflammatory markers and hypertension-mediated organ damage (HMOD). This was a cross-sectional study that included patients with primary arterial hypertension older than 18 years old. We included some parameters of 24 h ambulatory blood pressure monitoring collection and several inflammatory markers, as follows: platelet count (PTC), erythrocyte sedimentation rate (ESR), ultrasensitive C-reactive-protein, ferritin, fibrinogen, and uric acid. Myocardial hypertrophy, albuminuria, carotid intima-media thicknesses and ankle brachial index were assessed as HMOD presentations. Individuals were divided into two groups: patients with and without HMOD. We included 522 patients (47% women, mean age of 54 years). Multivariate logistic regression analysis showed that male patients older than 50 years old with uric acid levels above 7 mg/dL, ESR higher than 20 mm/h, fibrinogen greater than 320 mg/dL and PTC lower than 275 × 10^3^/µL were associated with HMOD (*p* < 0.05). The circadian BP profile (dipper versus non-dipper pattern) did reach neither statistical significance nor influence the odds ratio of those inflammatory markers for HMOD. We found that differences in some inflammatory markers between patients with and without HMOD were not explained by a different circadian BP profile.

## 1. Introduction

Elevated blood pressure (BP) is a major global contributor to cardiovascular (CV) risk and premature death. Both office and out-of-office BP have an independent relationship with the onset of several CV events [1,2].

The presence of hypertension-mediated organ damage (HMOD) as a surrogate marker of inadequate BP control is related to an increased CV risk and mortality, and its prevention should be a therapeutic target. The literature also supports that 24-h ambulatory BP monitoring (ABPM) is a better predictor of HMOD and CV events than office BP [3,4].

Recent evidence suggests that inflammation, immunity and arterial hypertension (AHT) may be related to each other, triggering an unfavorable inflammatory status that might increase BP and lead to HMOD and CV disease. Furthermore, several studies revealed that the inflammatory status could play a role in abnormalities of BP circadian rhythm, while other studies suggest that BP circadian profile could be related to HMOD [5,6,7].

There is a large amount of evidence in AHT pointing to an association between some inflammatory markers and HMOD, highlighting the role of erythrocyte sedimentation rate (ESR), ultrasensitive C-reactive protein (US-CRP), uric acid, ferritin and fibrinogen. Furthermore, several blood count parameters including white blood cell (WBC) count, platelet count (PTC), and medium platelet volume (MPV) have become universally available markers of chronic low-grade inflammation, and multiple studies have pointed to an association between these markers and atherosclerosis, distortion of circadian BP profile and HMOD [8,9,10].

However, it is less well known whether a dipper and non-dipper BP profile could influence this correlation between some inflammatory markers and HMOD. Thus, the aim of our study was to explore the relationship between some inflammatory markers and HMOD, and to assess whether this correlation could be independent of the circadian BP profile.

## 2. Materials and Methods

### 2.1. Study Design and Population

This was a cross-sectional study with a retrospective data analysis that was conducted at the Department of Internal Medicine (Hypertension and Cardiovascular Risk Unit) of the University Clinical Hospital of Santiago de Compostela (Galicia, Spain). Patient recruitment and data collection took place between January 2008 and December 2014.

The study included patients with primary AHT who were greater than 18 years old. Causes of secondary AHT were investigated in those patients with suggestive clinical findings according to recommendations proposed by the European Society of Cardiology (ESC) guidelines on AHT. Patients suffering from secondary AHT were not included [11].

Individuals with coronary arterial or cerebrovascular disease, acute or chronic renal impairment, fever, malignancy, or hematological disease were excluded from the study. Patients with acute or chronic respiratory diseases including obesity hypoventilation syndrome and sleep apnea syndrome were also excluded.

Suspected infection or any other inflammatory process was investigated on the basis of the anamnesis and physical examination. In case of clinical doubt, specific physical examinations and complementary tests were carried out. Patients suffering from any cause of acute or chronic inflammatory or infectious disease were not included.

HMOD presentations were considered on the basis of definitions that were supported by the European Society of Cardiology (ESC) guidelines on AHT as follows: electrocardiogram (ECG) voltage criteria for left ventricle hypertrophy (LVH) by Sokolow–Lyon criterion; albuminuria estimated with the albumin:creatinine ratio (ACR) in early morning spot urine sample, ankle–brachial index (ABI) measured with an automated device (Microlife WatchBP Office ABI ©) and carotid intima-media thickness (IMT) quantified by carotid ultrasound [11].

### 2.2. Parameters of Office BP and ABPM Collection

Patients underwent 24 h ABPM using an oscillometric Space-Labs 90207^®^ device (Space-Labs Inc., Redmon, WA, USA). Ambulatory BP was recorded every 20 min during the day (daytime: from 07:00 to 23:00) and every 30 min at night (night-time: from 23:00 to 7:00). The following indexes were available from the 24 h ABPM recordings: average systolic BP over 24 h (24-hSBP), average diastolic BP over 24 h (24-hDBP), average systolic BP during the day (dSBP), average diastolic BP during the day (dDBP), average systolic BP during the night (nSBP) and average diastolic BP during the night (nDBP), percentage of systolic BP fall during the night (SBPF), and percentage of diastolic BP fall during the night (DBPF). A non-dipper pattern was defined as a decrease in nSBP and/or nDBP < 10% compared with the average daytime values. A dipper pattern was defined as a decrease in nSBP and/or nDBP ≥ 10% and <20% compared with the average daytime values [12].

Quality controls were also performed to exclude poor 24 h ABPM data. Values of SBP < 70 or >250 mmHg, DBP < 40 or >150 mmHg, and heart rate (HR) < 40 or >150 bpm were excluded from the recording. The method was considered to be reliable if >70% of measurements were valid. Our study was conducted in accordance with the recommendations of the main international clinical practice guidelines on ABPM, and patients completed a diary about their sleep quality [13,14].

### 2.3. Clinical and Laboratory Variables

All patients were assessed for demographic characteristics (age and sex) and CV risk factors, including smoking status (non-smokers versus current or former smokers), alcohol intake (no consumption versus consumption of any amount), body mass index (BMI) with height: weight ratio in kg/m^2^, waist circumference (WC) measured above the iliac crests in cm, diabetes mellitus (DM), hyperlipidemia (HLP), antihypertensive treatment, and lipid lowering therapy. AHT and HLP were defined in accordance with ESC Clinical Practice Guidelines. DM was considered in accordance with the American Diabetes Association guidelines (ADA). WC was measured using the same standardized tape measure in all cases. Therapeutic compliance was evaluated with the Morisky–Green questionnaire [11,15,16].

Blood samples were obtained at 08:00 following overnight fasting. The following analytical variables were collected: complete blood count and biochemistry parameters including fasting plasma glucose (FPG), creatinine, and uric acid, lipid profile (total and fractionated cholesterol, triglycerides), US-CRP, ESR, ferritin, and fibrinogen. On the basis of evidence supporting inflammation in the pathogenesis of AHT and a possible role of some previously noted parameters in its quantification, we selected PTC, ESR, fibrinogen, uric acid, US-CRP and ferritin as inflammatory markers [8,9,10].

As for HMOD quantification, the threshold for abnormality was set as considered by clinical practice guidelines: ECG LVH: Sv1 + Rv5 > 35 mm; carotid IMT > 0.9 mm; ABI < 0.9; and ACR ≥ 30 mg/g [11].

### 2.4. Treatment of Variables and Sample Size

Variables of interest were collected in accordance with the information that was provided by our regional digital health records (IANUS, Galician health service [SERGAS]). Non-categorical predictive variables with clinical and/or statistical relevance were recoded into dichotomous ones to improve the efficiency of multivariate analysis and to conduct interaction–confusion studies. The cut-off points for categories were calculated as the value of the independent variable that best classified patients in the group with and without HMOD using the chi^2^ association test [17].

Taking into account the 24 h ABPM parameters and definitions previously discussed, the circadian BP profile was expressed as a qualitative variable called presence of a non-dipper profile (No/Yes) [12].

As the rationale of the study was to perform a global assessment of differences in some inflammatory markers between patients with and without HMOD and taking into account the existence of missing data in some HMOD presentations, we recoded all HMOD presentations as a qualitative variable which was the presence or absence of any type of HMOD. The sample size was determined by the total number of recruited patients.

### 2.5. Compliance with Ethical Standards

This study was included in the framework of the MAPEC project (registered at www.clinicaltrials.gov (accessed on 21 February 2022), with identifier code NCT00295542) and it was approved by the Institutional Ethics Committee of the University Clinical Hospital of Santiago de Compostela. All subjects gave their informed consent for inclusion. The study was conducted in accordance with the Declaration of Helsinki.

### 2.6. Statistical Analysis

Statistical analysis was performed using SPSS 20.0 statistical software (SPSS Inc., Chicago, IL, USA). Univariate analysis was used to explore individual associations between some inflammatory markers and HMOD. Comparisons were made between patients with and without HMOD. The Kolmogorov–Smirnov test was used to determine whether continuous variables were normally distributed. Normally distributed variables were analyzed using the Student’s *t*-test while non-normally distributed variables were analyzed using the Mann–Whitney *U*-test. Results were shown as the mean ± standard deviation (SD) and median with interquartile range (IQR), respectively. A chi-square test was used for categorical variables to test differences between groups. Results were shown as number (n) and percentage (%) of patients. For those variables with more than 5% of missing records, a missing value analysis was performed.

Multivariate logistic regression analysis was used to assess the correlation between inflammatory markers and HMOD taking into account possible interaction and confounding phenomena between clinically relevant variables. The inclusion criterion for the variables was a *p*-value of lower than 0.05 (*p* < 0.05) in the univariate analysis. Variables with *p* ≥ 0.20 were not shown. Results were expressed as the odds ratio (OR) and 95% confidence interval (95%CI) for HMOD. Statistical power was calculated as a function of the sample size, confidence level and a minimum effect size to be detected for the inflammatory markers evaluated in patients with HMOD. Finally, we assessed the performance of some inflammatory markers and clinical variables for HMOD with a model of receiver operating characteristics (ROC) curves. Results were provided as the area under curve (AUC) and 95% CI. 

## 3. Results

### 3.1. General Clinical Characteristics

There were 522 patients (47% women) with a mean age of 54 years who were included in the study, and among them, 154 (29%) and 365 (69%) suffered from DM and HLP, respectively. A total of 232 (45%) patients showed a non-dipper BP profile while 211 (40%) individuals exhibited some type of HMOD. One out of three patients was a current or former smoker. There were clinical and statistical differences in age, sex, WC, presence of DM, HLP and a non-dipper profile between patients with and without HMOD (*p* < 0.05). The most relevant results are shown in Table 1 (All results are provided in Appendix A).

There was an increased prevalence of antihypertensive and lipid-lowering drugs use (statins) between patients with and without HMOD (*p* < 0.05). We observed more than 10% of missing values in therapeutic compliance that were shown to be missing at random (data not shown). Most patients were compliant, and there was no difference in therapeutic adherence between groups. Renin–angiotensin–aldosterone system (RAAS) blockers were the most widely used group of drugs, and angiotensin II receptor blockers (ARBs) were the most frequent molecules (Table 1 and Appendix A).

### 3.2. 24-h Ambulatory Blood Pressure Monitoring Indexes

As for the office BP measurement, patients with HMOD had higher levels of SBP than the control group. The 24 h ABPM showed that patients with HMOD had lower levels of 24-hDBP and dDBP than the control group (*p* < 0.05). The results are shown in Table 1 and Appendix A. Differences in LVH, albuminuria, ABI and carotid IMT are provided in Appendix A.

### 3.3. Laboratory Findings

Patients with HMOD had higher FPG and creatinine levels with lower total cholesterol (TC), low-density lipoprotein- cholesterol (LDL-C) and high-density lipoprotein- cholesterol (HDL-C) concentrations than individuals without HMOD (*p* < 0.05). ESR, uric acid and ferritin levels were higher in patients with HMOD with a greater PTC in the control group (*p* < 0.05). There were also relevant differences in fibrinogen levels. There were more than 50% missing values for the US-CRP. Table 2 shows the main laboratory findings.

### 3.4. Multivariate Analysis

After an adjustment by age, sex, smoking status, WC, DM, HLP, BP circadian profile, antihypertensive therapy and lipid-lowering drugs, the results of the multivariate logistic regression analysis showed that male hypertensive patients greater than 50 years old with uric acid levels above 7 mg/dL, ESR higher than 20 mm/h, fibrinogen greater than 320 mg/dL and PTC lower than 275 10^3^/µL were associated with HMOD (*p* < 0.05). The results of the multivariate analysis are shown in Table 3. Appendix A contains the multivariate logistic regression for HMOD with and without inclusion of the circadian BP profile. Figure 1 provides the performance of the model for HMOD including and excluding PTC, ESR, fibrinogen and uric acid.

## 4. Discussion

Our study is one of the first studies that focus on the influence that the circadian BP profile and some other variables related to CV risk could exert on the correlation between some inflammatory markers and HMOD. The main results were as follows: (1) a nondipper BP profile and HMOD showed a prevalence of 45% and 40% respectively; (2) univariate analysis showed that patients with and without HMOD had relevant differences in age, sex, smoking status, WC, DM, HLP, circadian BP profile, antihypertensive therapy and lipid-lowering drugs; (3) multivariate analysis supported that PTC, ESR, fibrinogen and uric acid were correlated with HMOD independently of the circadian BP profile and some other classical variables related to CV risk.

AHT may lead to deleterious structural and functional changes in heart, blood vessels, brain, kidney, retina, and other target organs. Subclinical HMOD is important as an intermediate stage in the continuum of CV disease and some inflammatory mechanisms seem to be crucial in the development of some AHT-related abnormalities such as HMOD and alterations in BP circadian profile [18,19].

There seems to be a dual interaction between inflammatory markers and HMOD but it remains unclear which one is more of a cause than a consequence. Several factors could be related to a pro-inflammatory environment and HMOD promoting this relationship. However, the vast majority of elements involved, the function they would play and the weight they would have in HMOD still represent a gap in evidence [20,21].

It is well known that HMOD is associated with endothelial dysfunction and an increasing number of studies have shown that some inflammatory markers are directly or indirectly associated with the presence of endothelial dysfunction through physical or chemical mechanisms, highlighting the role of platelets, ESR, fibrinogen and uric acid [22,23,24].

Uric acid levels are invariably associated with higher atherosclerosis burden and CV disease. The enzyme xanthine oxidoreductase has two dual isoforms. In situations of vascular damage such as AHT or myocardial ischemia the predominant isoform is xanthine oxidase which induces the formation of reactive oxygen species (ROS) contributing to endothelial dysfunction [25,26].

Some causes of organ injury are known to induce an increase in fibrinogen proteolysis leading to the release of fibrinogen-degradation products (FDPs). FDPs are able to interact with vascular endothelium enhancing an increase in vascular permeability. Inflamed endothelial cells can also call for circulating activated platelets, leading to platelet adhesion to the vascular wall. Upon adhesion, activated platelets release many pro-inflammatory cytokines and chemokines [27,28,29,30].

Some of the factors that could be involved in a higher uric acid and fibrinogen levels observed in patients with HMOD could also be related to a pro-inflammatory internal milieu leading to increased levels of some acute phase molecules that would elevate ESR as a known nonspecific marker of inflammation [31,32].

The literature also supports the existence of differences in inflammatory markers depending on age, sex, toxic habits, lipid profile, diabetes mellitus or abnormalities of the circadian BP profile, among others. Moreover, patients with HMOD are often individuals who share multiple CV risk factors, so one might suspect that the differences in levels of certain inflammatory markers between patients with and without HMOD could be attributable to clinical differences between groups [33].

However, our study suggests that the individual weight of some classical variables associated with HMOD on the levels of certain inflammatory markers could be negligible. In particular, the inclusion of the circadian BP profile within the model has not modified the correlations previously showed for PTC, ESR, fibrinogen and uric acid. These findings may support that multiple not-sufficient, not-necessary and also not-known factors could be behind the differences in some inflammatory markers between patients with and without HMOD.

### Study Limitations and Strengths

This was a cross-sectional study of real clinical practice in which we had to take into account the possibility of biases that were related to the design. Our study was carried out in Caucasian patients from the northwest region of Spain (Galicia) who exhibited similar features than people from the western European countries. Therefore, results must be interpreted with caution when applying them to other populations, races or ethnicities.

The consideration of only two circadian BP profiles (dipper and non-dipper) is a limitation to be highlighted as it leaves out patients with extreme dipper or riser BP profiles that, although they represent a minority, could provide different data and influence the results. [34].

It is widely known that the use of antihypertensive and lipid-lowering drugs and the presence of DM are more prevalent in patients with HMOD and in those individuals with abnormalities of the circadian BP profile. In this regard, our study also yielded substantial differences between groups [35]. 

Therefore, despite the efforts to manage confounders at the analysis phase, it is possible that the differences observed in inflammatory markers between patients with and without HMOD as well as the apparent better control of some diastolic BP indexes that was observed in patients with HMOD could be partially due to an unbalanced distribution of some relevant variables between groups.

The high number of missing values for US-CRP may have limited the existence of differences between patients with and without HMOD as supported by other studies [36,37].

Some types of HMOD were not evaluated, highlighting the presence of retinopathy or damage to the central nervous system. The design of a global model for HMOD lacks the explanatory power that individual assessment of the different types of HMOD presentation would have. 

However, we believe that one of the strengths of this study was the high statistical power achieved (close to 90%) which allow us to support the results in a more robust way. This study represents a first approximation that would need to be corroborated by better quality designs.

## 5. Conclusions

We found that differences in some inflammatory markers between patients with and without HMOD appear to be multifactorial and would not be explained only by individual CV risk factors including an abnormal circadian BP profile. The authors suggest that assessment of some inflammatory markers and blood count parameters should be included as a part of the evaluation and follow-up of hypertensive patients as an additional tool in the early suspicion of HMOD.

## Figures and Tables

**Figure 1 jcm-11-01252-f001:**
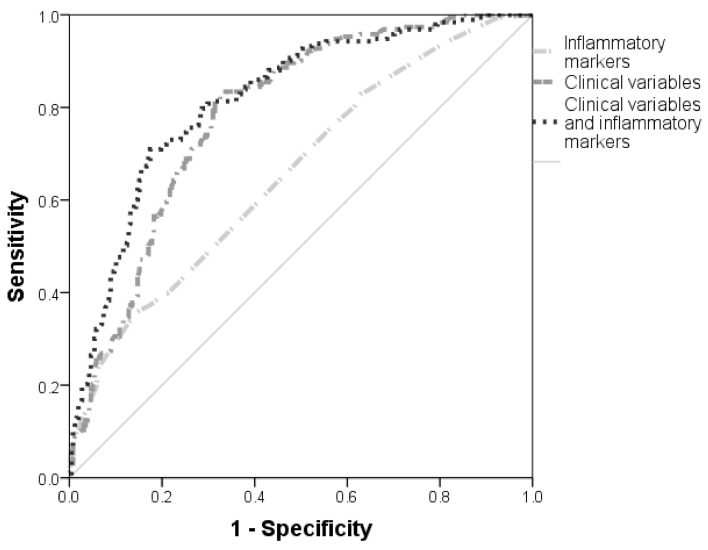
Receiver operating characteristics (ROC) curve of relevant clinical and laboratory variables for HMOD. Inflammatory markers (PTC < 275 10^3^µ/L, ESR > 20 mm/h, uric acid >7 mg/dL and fibrinogen >320 mg/dL): AUC 0.66 (0.03), *p* < 0.001, 95% CI 0.61–0.71. Clinical variables (age, sex, non-dipper profile, DM, ARABs: AUC 0.75 (0.021), *p* < 0.001, 95% CI 0.74–0.82. Clinical variables and inflammatory markers: AUC 0.82 (0.20), *p* < 0.001, 95% CI 0.77–0.86. Results were shown as area under curve (AUC) (Standard error), *p*-value, 95% confidence interval. PTC—Platelet count; ESR—Erythrocyte sedimentation rate; DM—Diabetes mellitus; ARBs—Angiotensin II receptor blockers.

**Table 1 jcm-11-01252-t001:** General features and 24 h ABPM parameters of patients with and without hypertension-mediated organ damage.

Variables	All Patients *n* = 522	Non-HMOD *n* = 311	HMOD *n* = 211
Age (years) †	54 ± 15	49 ± 14	62 ± 12 ^1^
Sex (female) ‡	247 (47)	169 (54)	78 (37) ^1^
WC (cm) †	100 ± 13	98 ± 13	104 ± 11 ^1^
Current/Former smokers ‡	153 (29)	82 (26)	71 (33) ^1^
Alcohol intake ‡	161 (30)	94 (30)	67 (31)
Nondipper profile ‡	232 (45)	112 (37)	120 (57) ^1^
HLP ‡	365 (69)	183 (58)	182 (86) ^1^
DM ‡	154 (29)	60 (19)	94 (44) ^1^
24-hSBP (mmHg) †	128 ± 13	127 ± 11	130 ± 14
24-hDBP (mmHg) †	76 ± 10	78 ± 10	74 ± 11^1^
dSBP (mmHg) †	133 ± 13	132 ± 12	134 ± 15
nSBP (mmHg) †	119 ± 14	117 ± 13	122 ± 15
dDBP (mmHg) †	81 ± 11	82 ± 10	78 ± 12 ^1^
nDBP (mmHg) †	68 ± 9	69 ± 9	67 ± 10
Antihypertensive drugs ‡	356 (68)	176 (56)	180 (85) ^1^
RAAS blockers ‡	259 (49)	117 (37)	142 (67) ^1^
ARBs ‡	225 (43)	93 (29)	132 (62) ^1^
Diuretics ‡	194 (37)	81 (26)	113 (53) ^1^
CCBs ‡	73 (23)	98 (46)	171 (32) ^1^
Statins ‡	174 (34)	76 (24)	98 (46) ^1^
Compliant patients ‡/§	338 (94)	171 (97)	167 (92)

HMOD—hypertension-mediated organ damage; WC—waist circumference; DM—diabetes mellitus; HLP—hyperlipidemia; SBP—systolic blood pressure; DBP—diastolic blood pressure; 24-hSBP—average SBP over 24 h; 24-hDBP—average diastolic BP over 24 h; dSBP—average SBP during the day; nSBP—average SBP during the night; dDBP—average DBP during the day; nDBP—average DBP during the night; RAAS—renin-angiotensin-aldosterone system; ARBs—angiotensin II receptor blockers; CCBs—calcium channel blockers. Results expressed as † refer to mean ± standard deviation, ‡ refers to number (%) and § refers to *n* = 356 patients. ^1^ Indicated comparison with patients without HMOD (*p* < 0.05).

**Table 2 jcm-11-01252-t002:** Laboratory finding in patients with and without HMOD.

Variables	All Patients	Non-HMOD	HMOD
FPG (mg/dL) †	110 ± 35	102 ± 24	122 ± 45 ^1^
Creatinine (mg/dL) †	0.94 ± 0.25	0.89 ± 0.2	1.00 ± 0.2 ^1^
TG (mg/dL) †	128 ± 81	122 ± 87	135 ± 71
TC (mg/dL) †	199 ± 41	202 ± 43	193 ± 39 ^1^
LDL-C (mg/dL) †	122 ± 36	126 ± 35	117 ± 36 ^1^
HDL-C (mg/dL) †	47 ± 14	48 ± 15	45 ± 13 ^1^
Uric Acid (mg/dL) ‡	5.7 (4.7–6.7)	5.5 (4.4–6.6)	6.1 (5.0–7.3) ^1^
ESR (mm/h) ‡	10 (5–19)	9 (5–17)	11 (6–22) ^1^
US-CRP (mg/L) †	0.89 ± 2.65	0.86 ± 2.33	0.95 ± 3.08
Fibrinogen (mg/dL) †	383 ± 87	378 ± 87	388 ± 92
Ferritin (µg/L) †	161 ± 124	150 ± 122	176 ± 129 ^1^
PTC (10^3^/µL) †	249 ± 60	255 ± 61	239 ± 59 ^1^
MPV (fL) †	9.0 ± 1.1	9.0 ± 1.0	9.1 ± 1.0
WBC count (10^3^/µL) †	7.39 ± 2.0	7.28 ± 1.9	7.55 ± 2.0
Hemoglobin (g/dL) †	14.5 ± 5.2	14.5 ± 6.6	14.3 ± 1.5

HMOD—hypertension-mediated organ damage; FPG—gasting plasma glucose; TG—triglyceride; TC—total cholesterol; LDL-C—low-density lipoprotein cholesterol; HDL-C—high-density lipoprotein cholesterol; ESR—erythrocyte sedimentation rate; US-CRP—ultrasensitive C–reactive protein; PTC—platelet count; MPV—medium platelet volume WBC count—white blood cell count. Results expressed as † refer to mean ± standard deviation and ‡ refer to median (interquartile range). ^1^ Indicated comparison with patients without HMOD (*p* < 0.05).

**Table 3 jcm-11-01252-t003:** Multivariate logistic regression of some clinical and laboratory variables for HMOD.

Variables	*p*-Value	Odds Ratio (OR) for HMOD	CI 95%	
Age (>50 years)	<0.001	3.628	2.078	6.337
Sex (female)	0.052	0.577	0.331	1.005
Non-dipper profile	0.171	1.387	0.868	2.215
DM	0.072	1.656	0.956	2.866
ARBs	0.021	2.133	1.120	4.062
PTC (<275 10^3^/µL)	0.011	2.010	1.176	3.434
ESR (>20 mm/h)	0.044	1.775	1.015	3.103
Fibrinogen (>320 mg/dL)	0.001	2.882	1.573	5.277
Uric acid (>7 mg/dL)	0.048	1.806	1.005	3.248

Omnibus test for coefficients (Chi^2^): *p* < 0.05. Cox and Snell R-squared: 0.281. Nagelkerke’s R-squared: 0.378. Sensitivity = 0.72. Specificity = 0.79. DM—diabetes mellitus; ARBs—angiotensin II receptor blockers; PTC—platelet count; ESR—erythrocyte sedimentation rate.

## Data Availability

Not applicable.

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
