# Peer review of "Are Differences in Inflammatory Markers between Patients with and without Hypertension-Mediated Organ Damage Influenced by Circadian Blood Pressure Abnormalities?"

_jcm, 2022, doi:10.3390/jcm11051252_

Round 1

Reviewer 1 Report

The original research entitled "Are differences in inflammatory markers between patients with and without hypertension-mediated organ damage influenced by circadian blood pressure abnormalities? is a well written and interesting draft, which evaluated some well-known aspects like the relationship between inflammatory biomarkers and HMOD, and other novel aspects like the potential effect of the modulation of those biomarkers by ambulatory hypertension phenotypes and HMOD development. Some minor aspects should be evaluated:

  • A treatment statistically significant imbalance between HMOD and no HMOD groups is observed. This data should be discussed regarding the impact on HMOD development and ABPM phenotypes
  • In the same scenario, and with the same recommendation, the frequency of diabetes is quite different in both groups. Diabetes is a well-known disease associated to higher frequency of non-dipping BP and higher frequency of HMOD. These huge and clinically relevant imbalances could be resolved by multivariate analysis?
  • The authors should discuss the ir/relevance of sample size in the absence of significant impact of circadian blood pressure profile, statistical type 2 error, reduced sample size?

Reviewer 2 Report

Very nice paper including a significant number of subjects. I have only one comment to the statistical approach. The authors conclude that their hypothesis was not supported by their study. Therefore, in the methods section it should be shown that the study has a statistical power to confirm these negative findings.
